# Dynamics of agonist-evoked opioid receptor activation revealed by FRET- and BRET-based opioid receptor conformation sensors
Sina B. Kirchhofer ⑩ , Claudia Kurz, Lorenz Geier, Anna-Lena Krett, Cornelius Krasel ⑩ & Moritz Bünemann ⑩ ✉

The opioid receptor family, particularly the μ opioid receptor, are the main drug targets in the management of severe pain. However, their pain-relieving effects are often accompanied by severe adverse effects, underlining the necessity for extensive research on this receptor family. Opioids, the agonists targeting these receptors, differ in their chemical structure and also in their mode of action in different aspects of signaling. Here we introduce novel tools that facilitate the analysis of this receptor family, by the development of FRET- and BRET-based receptor conformation sensors. With these sensors we were able to characterize especially the μ opioid receptor in more detail and reveal a strongly agonist-dependent activation kinetics for this receptor. Moreover, our sensors offer an assay independent from other signaling pathways, thereby minimizing the potential for interfering influences or biases within the system.

The family of opioid receptors plays a crucial role in pain relief, as they are the targets for the opioids, peptides that occur naturally or are created semi-synthetically or synthetically and used in pain management. The family consists of four members, the μ (MOR), kappa (KOR), delta (DOR) opioid receptor, and the nociceptin/orphanin FQ receptor (NOP), all transmitting their downstream signals via Gi/o proteins. The MOR is particularly important, being the main target for analgesics in the treatment of acute and severe pain. However, the analgetic effects of the opioid agonists are accompanied by severe side effects, like respiratory depression and addiction. Especially the outbreak of the recent opioid crisis in the US showed that the application of opioids is accompanied by a high risk for abuse and overdose, with more than 80.000 deaths caused by opioid overdose in 2021 alone[1]. There is a wide range of currently applied opioid analgesics which differ in their chemical structure and their properties of receptor activation and subsequent effects. They vary in their potency, efficacy, and kinetics to activate Gi/o proteins or other downstream signaling effectors via the MOR. Clinically opioids are natural and semi-synthetic opioids, like morphine and buprenorphine, or synthetic drugs, like fentanyl and methadone. Especially these synthetic drugs, most prominently fentanyl, are responsible for nearly 88% the observed deaths[1]. A better understanding of the action of the different opioids on different aspects of signaling is necessary. Activation of opioid receptors induces the activation of Gi/o proteins, which leads to the

replacement of GDP by GTP at the Gα subunit, inducing a dissociation or rearrangement of the heterotrimeric G proteins. These then transmit further signals, like the inhibition of the adenylate cyclase, inducing a decrease in cAMP levels, or the activation of ion channels. Moreover, the C-terminus of the activated receptor gets phosphorylated by GRKs. This allows the recruitment of arrestins to the receptor, inducing internalization, degradation, or recycling of the receptors. It is known that the different opioids induce different effects on the different stages of signaling, shown for example by measurements of GTPγS binding vs. arrestin recruitment[2], analysis of agonist-selective phosphorylation levels of the receptor[3], or analysis of different downstream effectors like G protein activation, recruitment of nanobodies, cAMP inhibition or activation of GIRK channels[4]. Until now, to our knowledge, no FRET- (Förster resonance energy transfer) nor BRET (bioluminescence resonance energy transfer)-based reporter systems for the detection of opioid receptor conformational changes on the level of the receptor itself have been successfully applied except for purified MOR[5]. In our study, we took advantage of the FRET-based receptor conformation sensors[6–9]. These conformation sensors are constructed as intramolecular sensors with a donor and an acceptor fluorophore, in our case mTurquoise2 and sYFP, which are inserted into the third intracellular loop and the C-terminus of the receptor. The binding of an agonist to a class A GPCR induces a rearrangement of the receptor

Department of Pharmacology and Clinical Pharmacy, University of Marburg, Marburg, Germany. ✉e-mail: moritz.buenemann@staff.uni-marburg.de

conformation, resulting in an outward movement of the sixth transmembrane domain[10]. These activation-dependent changes of receptor conformation can be detected with the FRET-based approach. The readout of these sensors is independent of any other signaling pathway, and there are no influences on the outcome through downstream signaling or amplification of specific signaling pathways. With these kinds of sensors, it is also possible to analyze the real velocity of agonist-induced conformational changes within the receptor[6]. In this study, we show the development of FRET- and BRET-based receptor conformation sensors for the MOR as well as the closely related KOR, which are suited for single-cell measurements, revealing kinetics of receptor activation and deactivation, and for microtiter plate format, allowing for higher throughput. These receptor sensors have a truncated C terminus removing the phosphorylation sites in the C terminus which are required for arrestin interaction.

## Results

### Generation of FRET-based conformation sensors for the μ opioid receptor

Our objective was to develop a Förster resonance energy transfer (FRET)-based receptor conformation sensor targeting the μ opioid receptor (MOR). To achieve this, we have used a methodology analogous to that previously described by Kurz et al. (2020), which was originally designed for the Thromboxane A2 (TP) receptor. Accordingly, we introduced a yellow fluorescent protein variant (sYFP) within the third intracellular loop and a cyan fluorescent protein variant (mTurquoise2) at the truncated C-terminus of the rat MOR. As previous studies[11–13] showed that the development of a functional and reliable μ opioid receptor sensor is problematic, especially the correct expression of the sensor at the plasma membrane, we inserted the fluorophores at different positions in the third intracellular loop (Fig. 1A). The intracellular loop of the MOR consists out of five amino acids (R-M-L-S-G). As it is known that after activation of the receptor, the sixth helix displays an outward movement, we inserted the fluorophores solely after the amino acids closer to helix six and not to the first one (R263). Further, we truncated the C terminus of the MOR after position T370 or after position T364 (version called *short*). We transfected each construct into HEK293T cells (human embryonic kidney cells) and tested the plasma membrane localization as well as the agonist-induced

FRET change in single-cell recordings at a microscope. Insertion of the sYFP after position M264 led to a construct which was expressed mainly at the membrane. Application of the full peptide agonist DAMGO led to a decrease in FRET emission ratio. However, the induced FRET change was less than 5% (Fig. 1B and Supplementary Fig. 1A). The membrane localization of this construct could be highly increased by further shortening the C terminus, nevertheless, this resulted in a weaker FRET change and worse signal-to-noise ratio (Fig. 1B and Supplementary Fig. 1B). The constructs with the insertion site S266 (Supplementary Fig. 1E) and L265 short (Supplementary Fig. 1D) showed a relatively weak membrane expression as well as a weak FRET change (Fig. 1B). The membrane expression and the FRET change was increased to approximately 5% for the L265 construct (Fig. 1B and Supplementary Fig. 1C). The two constructs having the sYFP inserted after G267 showed the strongest agonist-induced change in FRET emission ratio with approximately 6% (Fig. 1B, D and Supplementary Fig. 1F), due to an increase in CFP- and a decrease of YFP-emission after agonist application (Supplementary Fig. 1G). Further, both constructs were expressed predominantly at the plasma membrane (Fig. 1C and Supplementary Fig. 1F). As the signal-to-noise ratio and the membrane expression were slightly better for the G267 construct (Fig. 1C, D) in comparison to the version with a further truncated C terminus (Fig. S1F), we decided to continue with this construct, hereafter called MOR FRET sensor.

### Characterization of newly established MOR sensors

Subsequently, we established a HEK293 cell line which was stably expressing the MOR FRET-based conformation sensor. With this cell line, we conducted FRET measurements in a 96-well microtiter plate configuration using a plate reader. We therefore applied increasing concentrations of DAMGO, followed by a saturating concentration of DAMGO for normalization and the antagonist naloxone (Fig. 2A). From the resulting dataset, we generated concentration–response curves for various opioid agonists (Fig. 2B and Supplementary Fig. 2A–E) and subsequently determined the corresponding half-maximal effective concentration ($EC_{50}$) values (Table 1). Furthermore, we conducted an inhibition curve to characterize the concentration-dependent inhibition of the MOR by naloxone (Fig. 2C and Supplementary Fig. 2F). To this end, we activated the receptor using a concentration of DAMGO app. around the $EC_{50}$ and applied

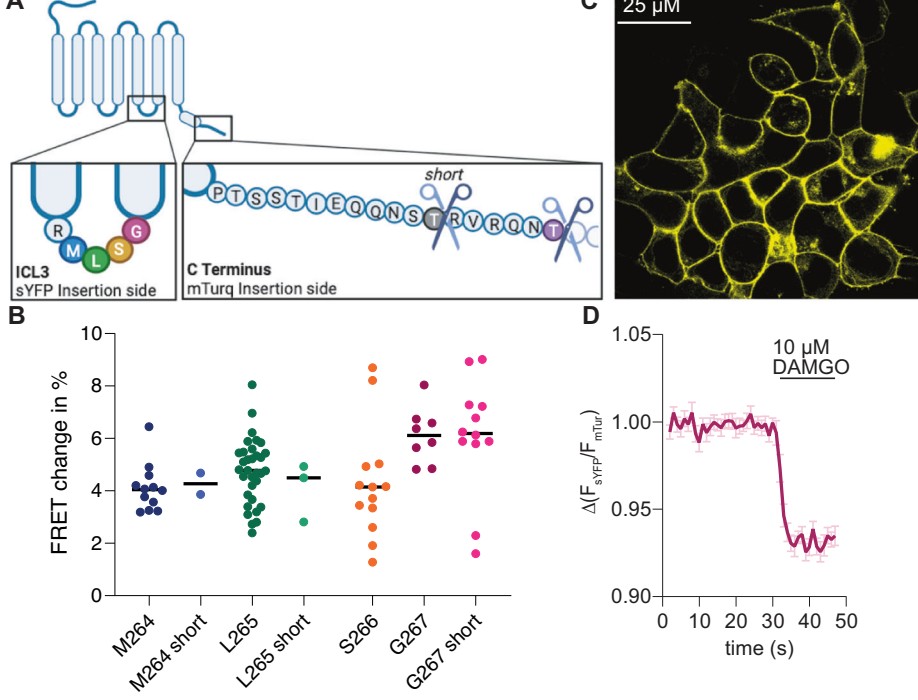

**Fig. 1 | Generation of MOR FRET-based conformation sensor. A** Schematic overview of the rat MOR. Different sites were chosen to insert a sYFP into the intracellular loop 3, after M264 (blue), L265 (green), S266 (red) or G267 (magenta). The C-terminus was shortened after T370 (violet) or T364 (gray, short), and an mTurq2 was appended. Scheme created with BioRender (Agreement Number: JP270G30ZL). **B** Scatter dot-plot showing the relative change of the FRET emission ratio induced by agonist application on the different MOR sensor constructs. **C** Representative picture of HEK293 cells stably expressing the MOR FRET sensor G267 in the YFP channel. **D** Averaged FRET-based single-cell recording of the MOR conformation sensor with sYFP inserted after G267 and mTurq2 inserted after T370. Application of DAMGO induced a robust decrease in FRET emission ratio by about 6% (mean ± SEM; n = 8).

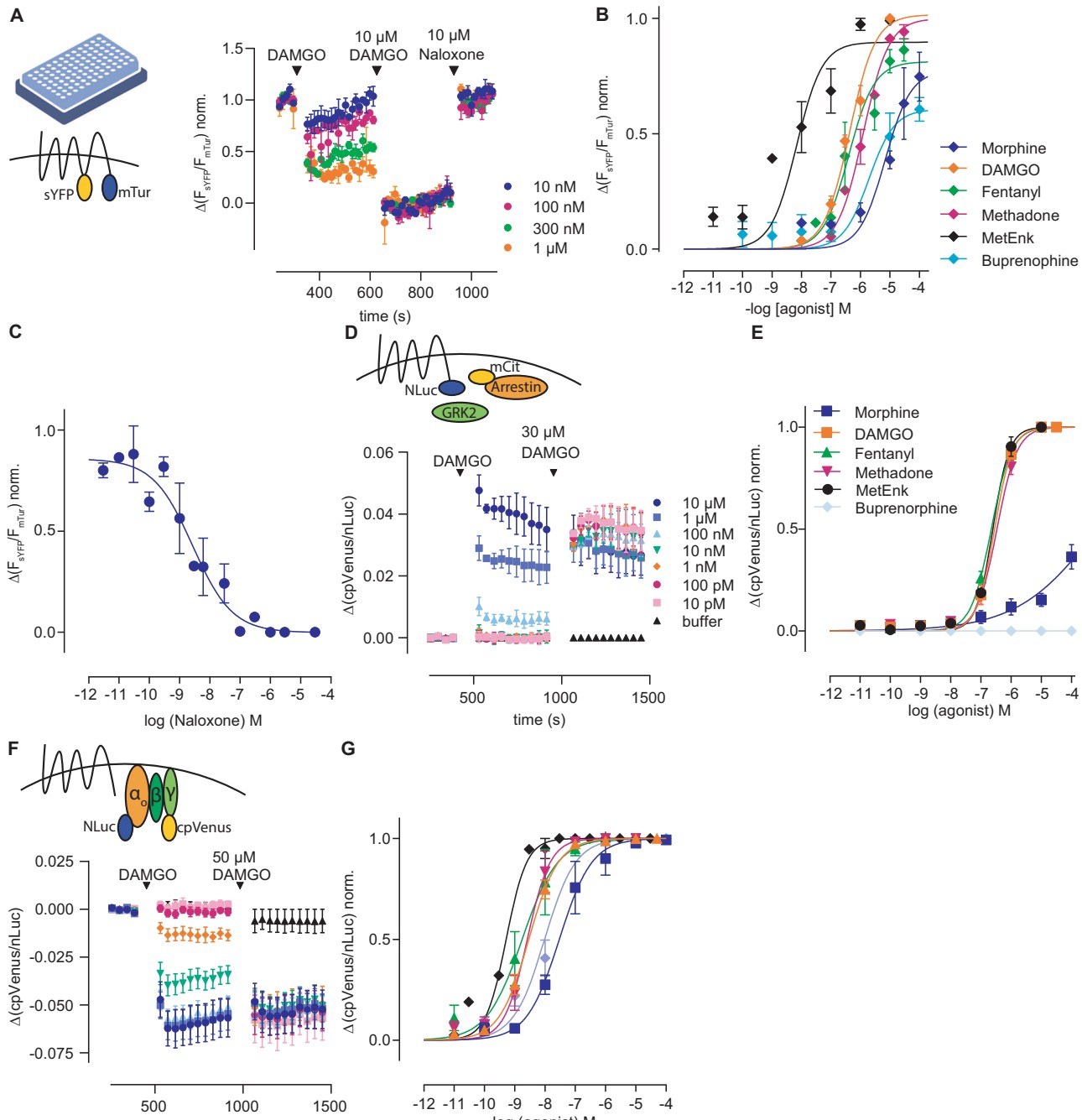

**Fig. 2 | Characterization of MOR FRET-based receptor conformation sensor.**
**A** Averaged FRET-based measurement of HEK293 cells stably expressing MOR conformation sensor measured in a 96-well plate format (schematic image of a 96-well plate created with BioRender, licence INC-12457) at a plate reader. Increasing concentrations of DAMGO were applied followed by a saturating concentration of DAMGO and the antagonist naloxone. The agonist-induced activation was normalized to the maximal activation and plotted as concentration–response curve (**B**) (mean ± SEM, $n = 3$ of independent transfections measured in triplets).
**B**, **C** Concentration–response (**B**) and inhibition (**C**) curves for the FRET MOR sensor (IC$_{50}$: 2.7 nM; pKi: 1.1 nM). **D**, **E** Averaged BRET-based measurement (**D**) of

HEK293T cells expressing MOR-nLuc, mCit-Arrestin3 and GRK2. Increasing concentrations of DAMGO were applied followed by a saturating concentration of DAMGO. The agonist-induced activation was normalized to the max. activation and plotted as concentration–response curve (**E**) (mean ± SEM, $n = 3$ of independent transfections measured in triplets). **F**, **G** Averaged BRET-based measurement (**F**) of HEK293T cells expressing MOR and Go-case BRET-sensor. Increasing concentrations of DAMGO were applied followed by a saturating concentration of DAMGO. The agonist-induced activation was normalized to the max. activation and plotted as concentration–response curve (**G**) (mean ± SEM, $n = 3$ of independent transfections measured in triplets).

increasing concentrations of naloxone. We then calculated the IC$_{50}$ for naloxone with 2.7 nM and the Ki with 1.1 nM, in line with the literature[14]. To compare our calculated EC$_{50}$ values for the receptor sensor, we performed BRET-based experiments on different MOR-induced signaling pathways. Therefore, we measured agonist-induced arrestin3 interaction with the

receptor. For this, HEK293T cells were transfected with MOR-nLuc, mCit-arrestin3 and GRK2. We applied increasing concentration of the respective agonist (Fig. 2D and Supplementary Fig. 2G–K), which induced an increase in the BRET ratio. We further applied a saturating concentration of DAMGO for normalization, fitted concentration-response curves (Fig. 2E),

and calculated $EC_{50}$-values for each agonist (Table 1). The $EC_{50}$-values of the sensor were ~1–5 times higher than the ones for receptor-arrestin interaction. Excluded were the partial agonists morphine and buprenorphine, which caused only weak or no receptor-arrestin interaction. As shown before, buprenorphine was not able to recruit arrestin to the MOR at all[2]. As the sensor is presumably uncoupled of the G proteins, as described for other GPCR FRET sensors, the receptor is likely not in the high-affinity state, which explains the shift of the $EC_{50}$-values[6]. We further performed BRET-based assays to measure the G protein activation induced by MOR agonists. We therefore used a BRET-based activity sensor Go1-case[15] in the same manner as for receptor-arrestin interaction (Fig. 2F and Supplementary Fig. 2L–P), fitted concentration-response curves (Fig. 2G) and calculated the $EC_{50}$-values (Table 1). The $EC_{50}$ values for MOR-induced G protein activation were all in the low nM range and strongly left-shifted in comparison to the sensor. As one receptor can activate more than one heterotrimeric G protein, there is a strong amplification of the signal, shifting the $EC_{50}$ values. Overall, these results suggest that the newly constructed FRET-based receptor conformation sensor for the MOR is detecting various opioids in a physiological range and is suitable for analysis in high-throughput setting in microtiter plate format.

As we were able to establish a FRET-based receptor conformation sensor, we further aimed to expand the field of application for this conformation sensor by creating a comparable BRET-based sensor. For this, we replaced the MOR FRET sensor, the mTurq2 at the truncated C terminus by a NLuc2, a Nano Luciferase. Again, we generated a HEK293 cell line stably expressing this conformation sensor. Generation of a concentration-response curve for DAMGO (Fig. 3A) resulted in converging curves for the FRET- and BRET-based sensors (Fig. 3B). Further, the calculated $EC_{50}$ values were highly comparable ($EC_{50}$ values of 445 nM (BRET) and 430 nM (FRET), Fig. 3C). Furthermore, the BRET-sensor was sensing fentanyl and morphine (Supplementary Fig. 3A). After application of a non-saturating

concentration of DAMGO, the signal was reversible by application of the antagonist naloxone (Supplementary Fig. 3B) and application of extracellular buffer induced no change in BRET ratio (Supplementary Fig. 3C).

## Ligand-specific activation kinetics revealed by the MOR FRET-based receptor conformation sensor

Activation and deactivation of GPCRs are mostly measured in downstream signaling, indirect or amplified assays. These assays do not reflect the velocity of the activation of the receptor itself. To gain information on the velocity of activation-induced conformational changes of the MOR by different agonists, we performed single-cell measurements of the MOR FRET sensor at a microscope. At the microscope, it was possible to measure with a high frequency of 10 Hz. Further, the microscope was combined with a pressurized perfusion system, allowing the fast change of solutions. This made it possible to analyze the on-and-off kinetics of the different opioid agonists. To do so, a full saturating concentration of 100 μM of the respective agonist was applied (Fig. 4A). After a plateau for activation was reached, the agonist was replaced again by an extracellular buffer (Fig. 4B). We performed a one-phase mono-exponential fit and calculated the half-time of activation (Fig. 4C and Supplementary Fig. 4A–F) and half-time of deactivation (Fig. 4D) for the respective agonist. In case of the peptides DAMGO and Met-enkephalin a mono-exponential curve fitted only to the initial phase of the curve (Supplementary Fig. 4A, B), which revealed the shortest half-time of activation compared to the other agonists. The peptides DAMGO and Met-enkephalin displayed a two-phase kinetics, probably due to a more complex binding mode of these peptides within the binding pocket of the MOR. For better comparability between the different agonist, we here only compare the first activation phase. Morphine and buprenorphine displayed the longest half-time of activation with ~7–8 s (Fig. 4C). However, we noticed that there is an obvious delay of the receptor-activation after the application of buprenorphine. This cannot be attributed to its moderate affinity to the receptor, since it is between fentanyl and morphine. Therefore, the lower efficacy of buprenorphine compared to the other agonists might be responsible for this effect. Indeed, as depicted in Fig. 2B, buprenorphine (and to a lesser extent morphine) does not induce the maximal FRET alteration compared to the other agonists, indicating an incomplete rearrangement of the helices. The half-time of deactivation was less than 10 s for DAMGO, Met-enkephalin, fentanyl, and morphine (Fig. 4D). Methadone and buprenorphine displayed a half-time of deactivation of more than 20 s (Fig. 4D). We further calculated the off-rate $K_{off}$ (Fig. 4E), giving the agonist-specific kinetic or speed od deactivation. The fastest agonist-washout was detected for fentanyl, followed by DAMGO. Again, methadone and buprenorphine displayed the slowest off-rate. Based on the $K_{off}$ and the respective $EC_{50}$ value, we calculated the on-rate $K_{on}$ for

**Table 1 | $EC_{50}$ values for the FRET sensor, arrestin interaction, and Go activation from Fig. 2**

| $EC_{50}$ values | FRET sensor | arrestin3 interaction | Go activation |
|---|---|---|---|
| DAMGO | 430 nM | 282 nM | 2.92 nM |
| Morphine | 6.93 μM | 713 μM | 29.7 nM |
| Fentanyl | 400 nM | 221 nM | 1.76 nM |
| Methadone | 1.23 μM | 339 nM | 2.61 nM |
| Met-enkephalin | 6.83 nM | 248 nM | 524 pM |
| Buprenorphine | 1.90 μM | n/a | 9.72 nM |

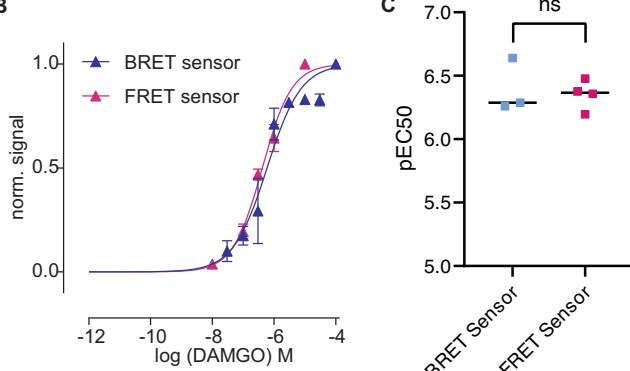

**Fig. 3 | Characterization of MOR BRET-based receptor conformation sensor. A** Averaged BRET-based measurement of HEK293 cells stably expressing MOR BRET-sensor. Increasing concentrations of DAMGO were applied followed by a saturating concentration of DAMGO. The agonist-induced activation was normalized to the maximal activation and plotted as concentration-response curve (**B**)

(mean ± SEM, $n = 3$ of independent measurements performed in triplets). **B** Concentration–response curve for DAMGO at the MOR BRET (blue) and MOR FRET (magenta) receptor conformation sensor. **C** Calculated $EC_{50}$ values for DAMGO (**B**) were comparable between BRET (445 nM) and FRET-based sensor (430 nM) ($P = 0.348$, unpaired $t$ test with Welch's correction).

**Fig. 4 | Ligand-specific activation kinetics of the MOR. A**, **B** Averaged single-cell FRET-based measurement of HEK293 cells stably expressing MOR conformation sensor. Cells were superfused with 100 µM of the indicated agonist, and measurements were recorded with a frequency of 10 Hz (**A**). After reaching a plateau, the applied agonist was washed off with extracellular buffer (indicated with the arrow) (**B**). **C**, **D** Half-time of activation (**C**) or deactivation (**D**) were evaluated by a one-phase mono-exponential fit based on (**A**, **B**) and Supplementary Fig. 4A–F (mean ± SEM, significance calculated against DAMGO, Brown–Forsyth & Welch's ANOVA, $P$ values: MetEnk C: 0.4637, D: 0.4374; Fentanyl C: <0.0001, D: 0.3526; Methadone C: 0.0056 D: 0.0006; Morphine C: <0.0001 D: 0.0174; Buprenorphine C: 0.0021 D: 0.1026). **E** Kinetics of deactivation were analyzed by a one-phase mono-exponential fit (mean ± SEM, significance calculated against DAMGO, Brown–Forsyth & Welch's ANOVA, $P$ values: MetEnk 0.0115; Fentanyl 0.1796; Methadone 0.0004; Morphine 0.0139; Buprenorphine 0.0006). **F** Kinetics of activation was calculated based on the kinetics of deactivation (E) and the $EC_{50}$ of the respective agonist (Table 1); (mean ± SEM, significance calculated against DAMGO, Brown–Forsyth & Welch's ANOVA, $P$ values: MetEnk 0.4283; Fentanyl 0.0227; Methadone, Morphine, Buprenorphine: <0.0001). **G** The calculated $K_{on}$ was plotted against the affinity of the respective agonist indicated by the pEC50 obtained in Fig. 2B.

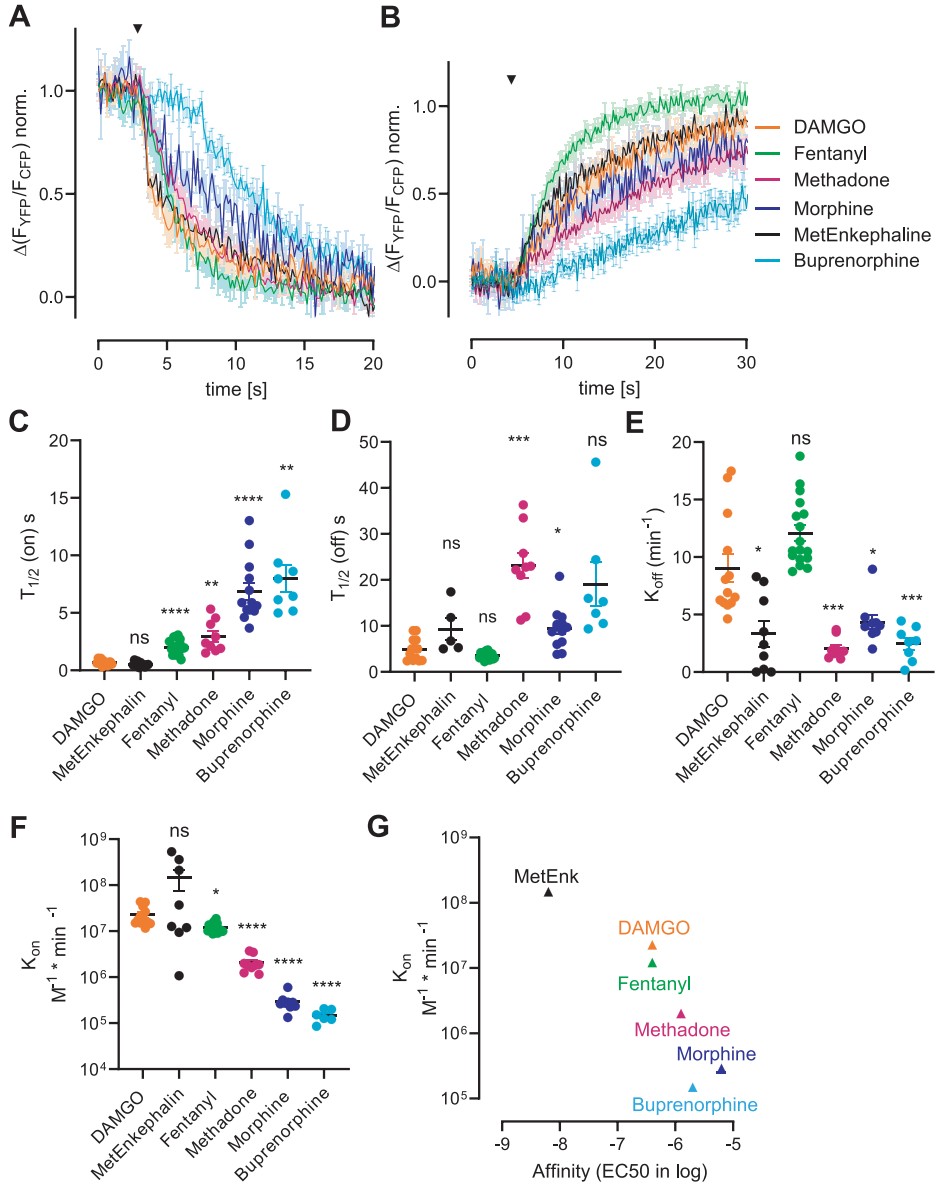

each agonist. The larger peptide agonists DAMGO and Met-enkephalin, as well as the smaller fentanyl, were characterized by a fast on-rate with approximately $10^7 M^{-1}*min^{-1}$ (Fig. 4F), which is comparable to kinetics calculated for other GPCRs like the $\alpha_{2A}$ adrenoceptor[6,16] or for prostaglandin receptors[17]. Methadone exhibited a slower on-rate which was in the range of $10^6 M^{-1}*min^{-1}$ (Fig. 4F), similar to the previously derived on-rate observed for the PTH receptor[6]. The partial agonists morphine and buprenorphine were substantially slower in their ability to activate the receptor, with a on rate of app. $10^5 M^{-1}*min^{-1}$ (Fig. 4F). The on-rate correlated with the respective affinity (Fig. 4G) and to some extent also with the respective efficacy of the agonist (Fig. 2B and Supplementary Fig. 4G). The agonists with the highest affinity like Met-enkephalin induced the fastest activation, whereas the partial agonists with the lowest affinity and efficacy like morphine and buprenorphine were considerably slower. The direct correlation between affinity (and to some extent efficacy) and speed could not be found for the off-rates of the agonists at the receptor. In this case, fentanyl showed the fastest off-rate (Fig. 4F). Met-enkephalin, morphine, methadone and buprenorphine were substantially slower. Taken together, the speed of activation of the MOR is correlated best with the affinity of the respective agonist, whereas the speed of deactivation interestingly did not show a close

correlation with agonist affinity or efficacy. Due to this unexpected finding, we conducted a comparative analysis of receptor activation kinetics using our novel FRET-based sensor and an alternative FRET-based assay detecting the interaction between the receptor and G-proteins. For this, HEK293T cells were transfected with MORsYFP, Gαo, Gβ and mTurq-Gγ. Application of DAMGO (Supplementary Fig. 4H) or morphine (Supplementary Fig. 4I) resulted in a rapid increase in FRET-emission ratio, induced by the interaction of the receptor with the heterotrimeric G-proteins. We calculated the half-time of activation for this interaction, indicating consistency between DAMGO-induced activation observed via the receptor sensor and that facilitated through the receptor-G-protein interaction, showing agreement between the two methodologies (Supplementary Fig. 4J). However, the receptor-G-protein interaction was notably faster for morphine than the sensor activation (Supplementary Fig. 4K). Additionally, our aim was to compare the deactivation kinetics. Yet, it became evident that the dissociation of the receptor from the G-protein exhibited a notably prolonged washout period compared to that observed with the sensor (Supplementary Fig. 4K, L), leading to a slower half-time of deactivation (Supplementary Fig. 4M) and overall deactivation kinetics (Supplementary Fig. 4N). As the activation kinetics were calculated based on

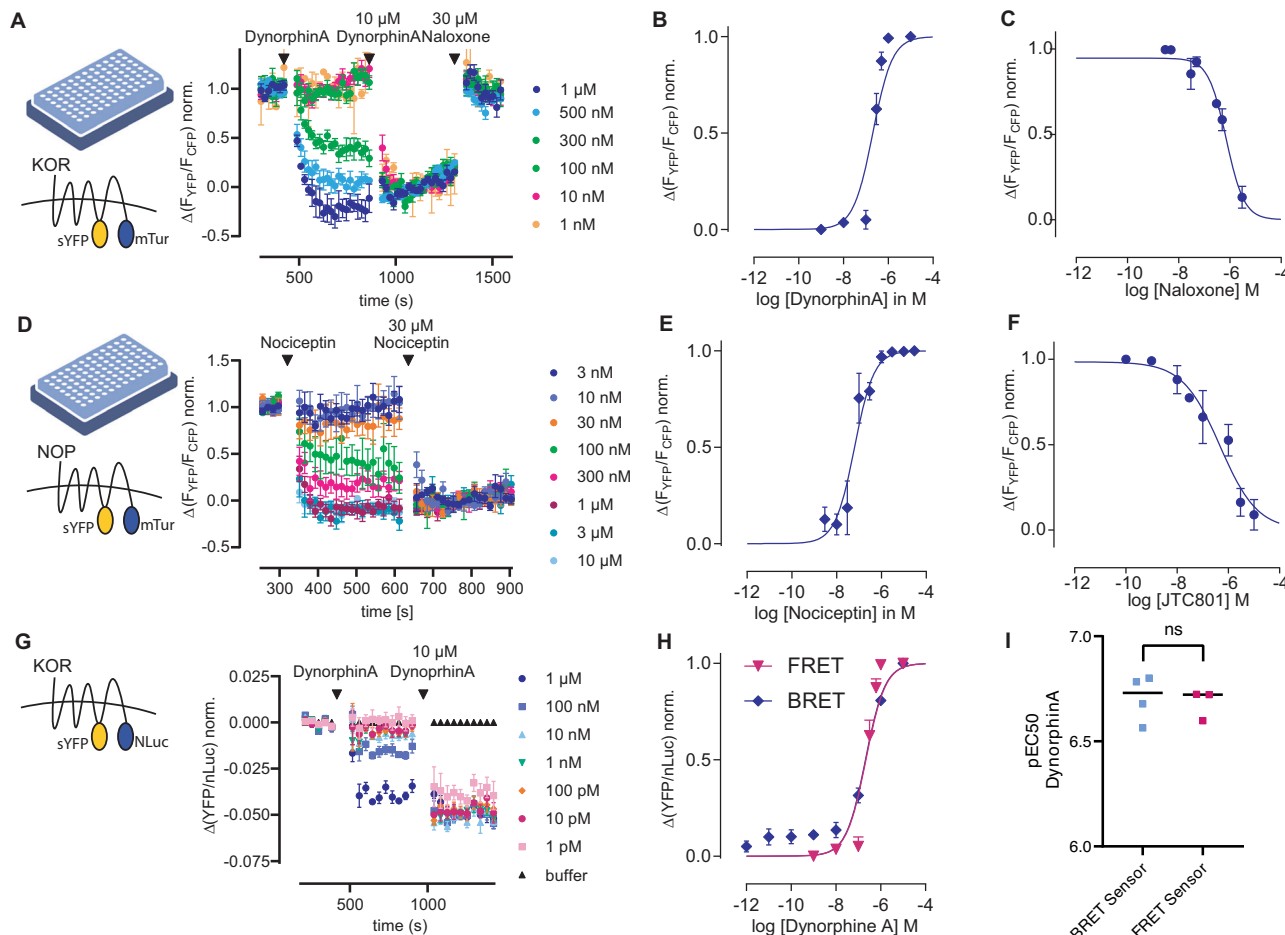

**Fig. 5 | FRET- and BRET-based sensors for human KOR and NOP. A, B** Averaged FRET-based measurement of HEK293 cells stably expressing the KOR conformation sensor measured in a 96-well plate format at a plate reader (schematic image of a 96-well plate created with BioRender, licence INC-12457). Increasing concentrations of dynorphine A were applied followed by a saturating concentration of dynorphine A and the antagonist naloxone. The agonist-induced activation was normalized to the maximal activation (mean ± SEM, $n = 5$ of independent transfections measured in triplets) and plotted as concentration-response curve for dynorphine A in (**B**) (EC$_{50}$: 219 nM). **C** Inhibition curve for the FRET KOR sensor activated with 500 nM dynorphine A and inhibited with increasing concentrations of the antagonist naloxone (IC$_{50}$: 716 nM; pKi: 314 nM). **D, E** Averaged FRET-based measurement of HEK293 cells stably expressing the NOP conformation sensor measured in a 96-well plate format at a plate reader (schematic image of a 96-well plate created with BioRender, licence INC-12457). Increasing concentrations of nociceptin were applied followed by a saturating concentration of nociceptin. The agonist-induced

activation was normalized to the maximal activation (mean ± SEM, $n = 3$ of independent transfections measured in triplets) and plotted as concentration-response curve for nociceptin in E (EC$_{50}$: 63 nM). **F** Inhibition curve for the FRET NOP sensor activated with 100 nM nociceptin and inhibited with increasing concentrations of the antagonist JTC-801 (IC$_{50}$: 473 nM; pKi: 298 nM). **G** Averaged BRET-based measurement of HEK293 cells stably expressing the KOR BRET-sensor. Increasing concentrations of dynorphine A were applied followed by a saturating concentration. The agonist-induced activation was normalized to the maximal activation and plotted as a concentration–response curve (**H**) (mean ± SEM, $n = 3$ of independent measurements performed in triplets). **H** Concentration–response curve for dynorphine A at the KOR BRET (blue) and KOR FRET (magenta) receptor conformation sensor. **I** Calculated EC$_{50}$ values for dynorphine A (H) were comparable between BRET (209 nM) and FRET-based sensor (219 nM) ($P = 0.7294$, unpaired $t$ test with Welch's correction).

the off-rate, the kinetics for receptor:G-protein interaction were substantially decelerated in comparison with the sensor (DAMGO: 7× slower, Morphine: 2× slower, Supplementary Fig. 4O). Nevertheless, the kinetics displayed a general alignment in comparable ranges, reinforcing our findings that the full agonist DAMGO elicits activation of the MOR at a notably faster rate compared to the partial agonist morphine (sensor: 75× faster, receptor:G-protein interaction: 20x faster).

**Approach can be applied to related receptors**
Given the high functionality observed with both the rat MOR FRET- and BRET-based receptor conformation sensors across different approaches, we applied our strategy on the remaining family members of the opioid receptors. Sequence alignments of the family revealed the ICL3 was nearly identical (R-L/M-L-S-G) in all family members. As the insertion of sYFP after the G turned out to be the best insertion site, we chose this site for the

human kappa opioid receptor (KOR), human delta-opioid receptor (DOR), and human nociceptin/orphanin FQ receptor (NOP) as well. Further, we chose the identical truncation length for the C terminus and inserted the mTurqoiuse2 in the same manner as at the MOR. We measured HEK293T cells transfected with either KOR or NOP FRET sensor at a microscope. Both sensors were localized at the cell membrane and showed an agonist-induced FRET-change of about 6% (Supplementary Fig. 5). Further, the establishment of HEK293-cell lines stably expressing the respective FRET sensor allowed the characterization of each sensor by recording of concentration-response curves in 96-well plate format in a plate reader. This yielded an EC$_{50}$-value of 219 nM for the endogenous agonist dynorphine A at the KOR (Fig. 5A, B). Furthermore, the inhibition curve with naloxone resulted in an IC$_{50}$ of 716 nM and a Ki of 314 nM (Fig. 5C). Moreover, a concentration-response curve for the NOP stimulated with the endogenous agonist nociceptin let to an EC$_{50}$ of 63 nM (Fig. 5D, E). In addition, an

inhibition curve with the antagonist JTC-801 resulted in an $IC_{50}$ of 473 nM and a Ki of 298 nM, in agreement with the literature (Fig. 5F). As already performed for the MOR, we exchanged the mTurquoise2 with the nLuc, creating a BRET-based receptor conformation sensor. Once more, this approach proved successful for the KOR, demonstrating an agonist-dependent change in BRET ratio (Fig. 5G) and yielding highly comparable $EC_{50}$ values (Fig. 5H, I). However, our attempts to develop a fully functional BRET-based conformation sensor for the nociceptin receptor were unsuccessful. The sensor exhibited inexplicable behavior that deviated noticeably from the responses observed with the FRET-based sensor. For this reason, we did not further use this BRET-based receptor sensor. Moreover, our attempts to develop a sensor for the delta-opioid receptor, whether FRET- or BRET-based, proved unsuccessful. None of the constructs generated displayed adequate expression levels at the cell membrane.

Taken together, our methodology enabled the development of receptor conformation sensors for the μ opioid receptor (MOR), kappa opioid receptor (KOR), and nociceptin/orphanin FQ receptor (NOP), functional for both FRET-based and BRET-based measurements, for MOR and KOR, respectively. Moreover, the FRET sensors exhibited functionality within a multiwell plate format and further were compatible for single-cell recordings at a microscope to directly resolve the activation kinetics of the receptor.

## Discussion

The opioid receptor family represents important drug targets, particularly in the management of severe pain which is mainly treated through the stimulation of μ opioid receptors. Nonetheless, the desired analgesic effects mediated by these receptors are often accompanied by severe adverse effects, underlining the necessity for extensive research to elucidate their mechanism of action. In recent years, diverse studies on the molecular characterization of the receptor and the effect of different agonists have been published, like several crystal and cryo-EM structures[18–20], as well as other approaches like the expression of the receptor as water-soluble protein[21,22]. In this study, we introduce novel tools that facilitate the analysis of this receptor family. It would also be desirable to have an assay independent from downstream signaling pathways, thereby minimizing the potential for interfering influences or biases within the system.

To achieve this, we successfully have taken advantage of the principle of receptor conformation sensors, which were described before[6–9] by utilizing FRET- and BRET-based approaches. Except for a recent study employing single-molecule FRET to study MOR conformations using chemically labeled purified receptors[5], to our knowledge, no FRET- or BRET-based reporter system directly reporting conformational changes of receptors has been established for the opioid receptor family. Several attempts to create a MOR-based sensor have been made[6–9], for example, based on the well-established so-called dLight sensors for the dopamine D1 receptor[11]. Patriarchi and colleagues[11] inserted a circular permuted GFP in the third intracellular loop of the receptor. As it is known that the sixth transmembrane domain of class A GPCRs like the MOR performs an outward movement upon agonist binding[10], the fluorescence of the cpGFP changes upon activation. However, the main problem was the membrane expression of these MOR-based constructs[11]. Recently, further attempts were carried out to establish an enhanced cpGFP-based receptor sensor[12,13]. The MOR sensor M-SPOTIT2[23], however, needed a high pH (> pH 9) and is working best in fixed cells, making it unusable for measurements of activation kinetics in living cells. Rappleye and colleagues[13] developed a high-throughput engineering platform to create receptor sensors carrying a cpGFP. They were also able to create a MOR sensor, called μMASS[13]. This MOR sensor showed a good membrane expression and a strong fluorescence change. However, this sensor was only sufficiently activated by endogenous peptide opioids like Met-enkephalin or its analogs. There was just a weak fluorescence change observed for the clinically used opioids like morphine or fentanyl. In our study, we applied another, well-established approach: the FRET-based receptor conformation sensors. FRET-based receptor conformation sensors are intramolecular sensors consisting of one protein with the insertion of two different fluorophores and the signal is obtained via changes in distance or orientation of these two fluorophores. As the receptor is intracellularly fused with two large fluorophores in the region of the G-protein binding site, an association with the heterotrimeric G-proteins is highly unlikely. Further, in our approach the C terminus was truncated, removing most of the phosphorylation sites in the C terminus which are required for arrestin interaction. Therefore, these FRET sensors should act independently of any other signaling pathways[6] and with this provide a potentially unbiased approach. In contrast to the previously described MOR sensors, our FRET-based sensor is well expressed at the membrane, works at physiological pH, is activated by a wide range of different opioid agonists, and was able to reveal the real-time activation kinetics for the different opioids.

Moreover, we were able to directly apply our approach on the other family members, as we developed comparable FRET-based sensors for the kappa opioid receptor as well as for the nociceptin/orphanin FQ receptor. Unfortunately, we were not able to create a functional delta-opioid receptor sensor, as none of our constructs was sufficiently expressed at the cell membrane.

It is worth noting that we were able to not only successfully establish FRET-based receptor conformation sensors, but we were further able to directly apply the same arrangement in a BRET-based approach, making our sensors suitable for diverse ways of utilization—from single-cell measurements with the FRET-based sensor, allowing high spatial and temporal resolution, to microtiter plate-based FRET- or BRET-based measurements, allowing higher throughput.

With these kinds of sensors, it is also possible to analyze the real velocity of agonist-induced conformational changes within the receptor. Our MOR FRET sensor revealed the on-rate for different opioid agonists. For the peptide agonists Met-enkephalin and DAMGO the onset kinetics were fast, however they could not be fitted monoexponentially (Supplementary Fig. 4A, B). This might support the conclusion that more than one conformation is induced by these agonists as previously reported based on single molecule FRET assays in purified MOR[5]. Met-enkephalin, which displayed the highest affinity to activate the receptor, had the fastest on rate, which was also in the range of the activation kinetics shown for other GPCRs[6,17]. DAMGO and fentanyl displayed a slightly slower affinity for receptor activation and induced a slightly slower activation as well. However, lower affinity as well as lower efficacy agonists like morphine displayed a noticeably decelerated on-rate. This finding, together with the missing correlation between receptor deactivation kinetics and agonist affinity, indicates that for MOR ligands, the $k_{on}$ is distinct for each agonist and has an unusually high impact on the difference in affinity and to some extent efficacy of the agonist. Overall, this suggests there is a highly ligand-specific effect underlying the activation kinetics of the MOR, but further, no clear ligand-specific effect on the deactivation kinetic of the receptor. These diverse kinetics could be explained by the differential interactions the different opioid agonists form with the receptor[24]. Further, it was shown that the opioids induce diverse conformations within the activated MOR[18]. In particular, fentanyl and morphine display different receptor interactions[24] and a different binding mode[18] and with this probably also distinct active receptor conformations, which could explain the differences in activation speed. In the past, receptor conformation sensors have revealed different activation kinetics for different GPCRs, ranging from time constants for the onset of activation in the range of a few tens of milliseconds for receptors activated by small molecules to hundreds to thousands of milliseconds for peptide-binding receptors[6,7,9,25]. However, here we show that the on-rate is also highly dependent on the agonist used for activation of the receptor. Further, the properties of the agonist, like affinity to the receptor, and somehow efficacy for the receptor activation, determine the activation speed of the receptor. In conclusion, we here introduce a novel tool that could facilitate further, potentially unbiased, analysis of the opioid receptor family from single-cell analysis to multiwell format, providing information not only on affinity of drugs to the receptor, but also on their efficacy in receptor activation.

## Methods

### Plasmids

cDNAs for the rat μ opioid receptor (MOR-wt), MORsYFP2, Gβ$_1$-wt, GRK2-wt[26], PTX-insensitive Gαo[27], mTurq-Gγ2[28] and BRET-based activity sensor Go$_1$-case[15] have been described before. Human nociception/orphanin FQ receptor was purchased from Addgene (Watertown, Massachusetts, USA, plasmid #66463), human kappa opioid receptor (catalog number #OPRK100000) and human delta-opioid receptor (catalog number #OPRD100000) were purchased from the Missouri S&T cDNA Resource Center (cDNA.org, Bloomsburg, Pennsylvania, USA). MOR-nLuc was constructed by fusion of rat MOR-wt with a C-terminal nLuc from nLuc-Gγ2 (kind gift of N. Lambert, Augusta University, Georgia, USA) using the NEBuilder Hifi DNA assembly kit (New England Biolabs, Ipswich, Massachusetts, USA). The nLuc was enclosed by the restriction sides AgeI and EcoRV. The following primers were used (sequence 5' → 3'): MOR-BB-fwd: GCATTCTGGCGTAATCTAGATAACTGGGTCTCACACCATCTAG AGG; MOR-BB-rev: AGTGTGAAGACCATACCGGTGGGCAATGGA GCAGTTTCTGCCTCCAGATT; nLuc-Insert-fwd: CTGCTCCATTG CCCACCGGTATGGTCTTCACACTCGAAGATTTCGTTGG; nLuc-Insert-rev: GTGAGACCCAGTTATCTAGATTACGCCAGAATGCGTT CG. sYFP-Arrestin3 was constructed via two steps. First, a pcSYFP2-C was cloned amplifying the sYFP2 (mVenus (L68V)) ORF from pmVenus(L68V)-C1, which was a kind gift by Joachim Goedhart. The following primers were used (sequence 5' → 3'): fwd: AAAAAAAAGCTTATGGTG AGCAAGGGCGAGG; rev: AAAAAAGAATTCCTTGTACAGCTCGTC CATGCC. The resulting PCR product was cut with HindIII and EcoRI and cloned into pcDNA3 cut with the same enzymes. The sYFP2-Arr3 was cloned by amplifying the bovine Arrestin3[29] using PCR with the following primers (sequence 5' → 3'): fwd: AAAAAAGATATCAGATGGGGGA GAAACCCGG, REV: AAAAAAGCGGCCGCTTAACAGAACTGGT CGTCATAG. The resulting PCR product was cut with EcoRV and NotI and cloned into pcsYFP2-C cut with the same enzymes. The MOR-FRET-Sensors were constructed by using the NEBuilder Hifi DNA assembly kit (New England Biolabs) based on the rat MOR-wt, the sYFP2 from MOR-sYFP, and the pcDNA3.1+ vector, including a C-terminal mTur2 from EP4-mTur2[17]. The C-terminus of MOR was truncated after position T370 or T364, and the mTurq2 was attached afterward including the restriction sides AgeI and EcoRV. The sYFP2 was inserted at four different positions in the ICL3: after M264, after L265, after S266, and after G267, and was enclosed by the restriction sides SacII and HpaI. The following primers were used (sequence 5' → 3'): Sensor M264: fragment 1 fwd: GGAATTCACC ATGGACAGCAGCACC; fragment 1 rev: CCCTTGCTCACCAT CCGCGGCATGCGAACGCTCTTGAGT; YFP fwd: AGAGCGTTCG CATGCCGCGGATGGTGAGCAAGGGCGAG; YFP rev: TTGGAGCC CGATAGGTTAACCTTGTACAGCTCGTCCATGCC; fragment 2 fwd: ACGAGCTGTACAAGGTTAACCTATCGGGCTCCAAAGAAAGGA C; fragment 2 rev: TCGCCCTTGCTCACACCGGTAGTGTTCTGAC GGACTCGAGT; vector + mTurq fw: TCCGTCAGAACACTACCGGT GTGAGCAAGGG; vector + mTurq rev: TGCTGTCCATGGTGAA TTCCACCACACTGGA. Sensor M264 short: same as M264, besides fragment 2 rev: TCGCCCTTGCTCACACCGGTAGTGGAGTTTTGC TGTTCGATCG and vector + mTurq fw: AGCAAAACTCCACTACC GGTGTGAGCAAGGG. Sensor L265: same as M264 besides fragment 1 rev: CCCTTGCTCACCATCCGCGGTAGCATGCGAACGCTCTTGAG; YFP fw: GCGTTCGCATGCTACCGCGGATGGTGAGCAAGGGCGAG; YFP rev: TCTTTGGAGCCCGAGTTAACCTTGTACAGCTCGTCCAT GCC; fragment 2 fw: ACGAGCTGTACAAGGTTAACTCGGGCT CCAAAGAAAAGGACAG. Sensor L265 short: fragment 1 fw and vector +mTurq rev same as M264, fragment 2 rev and vector-mTurq fwd same as M264 short, fragment 1 rev, YFP fwd, YFP rev and fragment 2 fwd same as L265. Sensor S266: same as M264, besides fragment 1 rev: CCCTTGCTCACCATCCGCGGCGATAGCATGCGAACGCTC; YFP fwd: TTCGCATGCTATCGCCGCGGATGGTGAGCAAGGGCGAG; YFP rev: TTTTCTTTGGAGCCGTTAACCTTGTACAGCTCGTCCAT GCCG; fragment 2 fw: ACGAGCTGTACAAGGTTAACGGCTCCAAAG

AAAAGGACAGGAATCT. Sensor G267: same as M264, besides fragment 1 rev: CCCTTGCTCACCATCCGCGGGCCCGATAGCATGCGAACG; YFP fwd: GCATGCTATCGGGCCCGCGGATGGTGAGCAAGGGCG AGG; YFP rev: TCCTTTTCTTTGGAGTTAACCTTGTACAGCTCG TCCATGCC; fragment 2 fwd: ACGAGCTGTACAAGGTTAACTCC AAAGAAAAGGACAGGAATCTGC. Sensor G267 short: fragment 1 fwd and vector+mTurq rev same as M264. Fragment 1 rev, YFP fwd, YFP rev and fragment 2 fw same as G267. Fragment 2 rev and vector+mTurq fwd same as M264 short. The MOR-BRET-sensor was generated based on the G267-MOR-FRET-sensor with the mTurq2 replaced by nLuc2 using following primers: vector fwd: GTGCGAACGCATTCTGGCGTGATAT CTAGCTCGAGTCTAGAGGGC; vector rev: TCTTCGAGTGTGAAG ACCATACCGGTAGTGTTCTGACGGAC; nLuc fwd: CGTCAGAAC ACTACCGGTATGGTCTTCACACTCGAAGATTTCGTTGGGGACT; nLuc rev: CTAGACTCGAGCTAGATATCACGCCAGAATGCGTTCG. The KOR and NOP FRET-sensor were created in the same manner as the MOR, using the following primers for the KOR: vector+mTurq fwd: CTAGCAGAGTCCGAAATACAACCGGTGTGAGCAAGGG; vector +mTurq rev: ATCTGAATCGGGGATTCCATGGTGAATTCCACCAC ACTGGA; fragment 1 fwd: CCAGTGTGGTGGAATTCACCATGGAAT CCCCGATTCAGA; fragment 1 rev: CCCTTGCTCACCATCCGCGGG CCAGAAAGGAGCCGGACG; YFP fwd: GGCTCCTTTCTGGCCCGC GGATGGTGAGCAAGGGCGAGG; YFP rev: TCTTTCTCTCGGGAGT TAACCTTGTACAGCTCGTCCATGCC; fragment 2 fwd: ACGAGCTGT ACAAGGTTAACTCCCGAGAGAAAGATCGCAAC; fragment 2 rev: TCGCCCTTGCTCACACCGGTTGTATTTCGGACTCTGCTAGTGCT. For the NOP FRET sensor the following primers were used: vector+mTurq fwd: TCTCAGACAGAGTGAGGTCAACCGGTGTGAGCAAGGG; vector+mTurq rev: GCCGGAAACAGGGGCTCCATGGTGAATTCCACC ACACTGGA; fragment 1 fwd: CCAGTGTGGTGGAATTCACCATGGA GCCCCTGTTTCCG; fragment 1 rev: CCCTTGCTCACCATCCGCGG TCCACTCAGCAGTCTAACGCCTCG; YFP fwd: GTTAGACTGCTGA GTGGACCGCGGATGGTGAGCAAGGGCGAG; YFP rev: GGTCCTTC TCCCTGCTGTTAACCTTGTACAGCTCGTCCATGCC; fragment 2 fwd: GGACGAGCTGTACAAGGTTAACAGCAGGGAGAAGGACC GC; fragment 2 rev: CTCGCCCTTGCTCACACCGGTTGACCTCACT CTGTCTGAGACTTGTACA. The BRET sensors for KOR and NOP were generated analogously to the MOR BRET-sensor with the mTurq2 replaced by nLuc2 using the following primers: KOR BRET-sensor: vector fwd: GTGCGAACGCATTCTGGCGTGATATCTAGCTCGAGTCTAGAGG GCC; vector rev: AGTGTGAAGACCATACCGGTTGTATTTCGGACT CTGCTA; nLuc fwd: GAGTCCGAAATACAACCGGTATGGTCTTCAC ACTCGAAGATTTCGTTGG; nLuc rev: CTAGACTCGAGCTAGAT ATCACGCCAGAATGCGTTCG. For the NOP BRET senor the following primers were used: vector fwd: GTGCGAACGCATTCTGGCGTGAT ATCTAGCTCGAGTCTAGAGGGC; vector rev: TCTTCGAGTGTGAA GACCATACCGGTTGACCTCACTCTGTCT; nLuc fwd: ACAG AGTGAGGTCAACCGGTATGGTCTTCACACTCGAAGATTTCGT; nLuc rev: same as used for MOR-nLuc: Insert nLuc rev. All oligonucleotide primers used in this study were designed using SnapGene Viewer (GSL Biotech, San Diego, California, USA) and purchased at Eurofins Genomics (Ebersberg, Germany).

### Cell culture

All experiments in this study were carried out in HEK293T or HEK293 cells, which were a kind gift from the Lohse laboratory, University of Würzburg, Germany. Cells were cultured in DMEM (Dulbecco's Modified Eagle's Medium, 4.5 g L$^{-1}$ glucose) supplemented with 10% FCS, 2 mM L-glutamine, 100 U/ml penicillin and 0.1 mg/ml streptomycin or 0.4 mg/ml G418 for selection at 37°C and 5% CO$_2$. Cells were tested on a regular basis for mycoplasma. HEK293T cells were transiently transfected in 6 cm Ø dishes using Effectene Transfection Reagent according to the manufacturer's instructions (Qiagen, Hilden, Germany) 2 days before the measurement. For the screening of the different MOR-FRET-sensors, cells were transfected with 1 µg of the respective sensor-construct. A HEK293 cell line stably

expressing the G267-MOR-FRET-sensor and MOR-BRET-sensor was established and used for measurements of this sensor construct. For BRET-based measurements, HEK293T cells were transiently transfected 2 days before the measurement using PEI (polyethylenimine) reagent. For the measurement of G-Protein activity, cells were transfected with 1.5 µg of MOR wt and 1.5 µg of the BRET-activity sensor Go1-case (100 ng DNA/well). For the measurement of receptor-arrestin interaction, cells were transfected with 0.5 µg MOR-nLuc, 5 µg sYFP-Arrestin3, and 0.5 µg GRK2 (200 ng DNA/well). For the BRET-based measurements of the KOR or NOP sensor, the cells were transfected with 3 µg DNA (100 ng DNA/well). The mixing ratio of PEI to DNA was 3:1 with 1 mg/ml PEI. Per 1 µg DNA, 50 µl serum-free DMEM was added. The DNA-DMEM and PEI-DMEM mix were incubated for 5 min, then they were combined and further incubated for 10 min. The cells were counted and set to 300,000 cells per ml (30,000 cells/well), the DNA-PEI mix was added and seeded onto white sterile poly-L-lysine coated microplates and incubated for 48 h.

### Reagents
DMEM, FCS, penicillin/streptomycin, G418, L-glutamine, and trypsin-EDTA were purchased from Capricorn Scientific (Ebsdorfergrund, Germany). DAMGO acetate salt, Met-enkephalin acetate salt, buprenorphine-HCl, and fentanyl citrate were purchased from Sigma-Aldrich (St. Louis, MO, USA). Morphine hydrochloride was purchased from Merck (Darmstadt, Germany), naloxone-HCl and dynorphin A TFA salt were purchased from Cayman Chemical (Ann Arbor, Michigan, USA), L-methadone-HCl was purchased from Hoechst AG (Frankfurt, Germany), nociceptin (1–13) amide TFA was purchased from TargetMol (Boston, MA, USA) and JTC-801 was purchased from Tocris Bioscience (Bristol, United Kingdom). 6H-F-Colenterazine was a kind gift of Dr. Wibke Diederich (University of Marburg, Germany) and was used as published previously[30].

### Single-cell FRET measurements
Single-cell FRET measurements were performed at an inverted microscope as described before[31]. Single-cell FRET measurements describing the activation and deactivation kinetics were measured at an inverted microscope (Eclipse Ti2-E, Nikon, Japan) equipped with a 100x oil-immersion objective (Plan Apo 100x Lambda/1.45 Oil/0.17 WD 0.13, Nikon), LED light source (CoolLED pE4000, CoolLED, Andover, UK) and a high-performance camera (Prime 95B Scientific CMOS, Teledyne Photometrics, Tucson, Arizona, USA). During FRET measurements, CFP was excited with 435 nm at an intensity of 20% using an excitation filter (438/24; Semrock, Rochester, New York, USA) and a dichroic beam splitter (458; Semrock). Fluorescence emission of CFP and YFP was simultaneously collected side-by-side by a second beam splitter (488, Chroma, Bellow Falls, USA) and two emission filters (CFP: 474/27; Chroma and YFP: 544/23; Chroma). Cells were illuminated by short light flashes for 50 ms with a frequency of 10 Hz. Images were captured using NIS-Elements advanced research imaging software (Nikon). For the measurements, cells were plated on glass coverslips coated with poly-L-lysin and were continuously superfused with either external buffer (137 mM NaCl, 5.4 mM KCl, 2 mM CaCl$_2$, 1 mM MgCl$_2$, 10 mM HEPES, pH 7.3) or the respective agonist-solution using a pressurized fast-switching valve-controlled perfusion system (ALA Scientific) allowing a rapid change of solutions. The measurements were carried out at room temperature.

### FRET measurements of multiple cells in the plate reader
Measurements of cells stably expressing the MOR-FRET-sensor were performed in a plate reader (Spark 20 M, Tecan). For this, black 96-well polystyrene microplates with clear bottom (either purchased from Brand (Wertheim, Germany) or Greiner (Kremsmünster, Austria), depending on availability) were coated with poly-L-lysine. 120.000 cells per well were seeded the day before the measurement. The measurement was performed as described before[17]. In brief, the measurement was performed as the bottom measurement with excitation of the CFP at 430 nm and the emitted donor fluorescence (CFP) was recorded at 485 nm and the acceptor

fluorescence (YFP) at 535 nm, respectively. The measurement was controlled using the software SparkControl. After the measurement, the YFP/CFP emission ratio was calculated using Microsoft Excel.

### BRET measurements
BRET measurements were performed using a Spark 20 M (Tecan) plate reader. Brand 96-well polystyrene microplates were coated with poly-L-lysine. In total, 30,000 cells per well were counted and seeded during transfection. Before the measurement, the culture medium was removed, and the cells were washed with an external buffer. Subsequently, 80 µl of external buffer containing 1 µM 6H-F-Colenterazine[30] was added into each well and incubated for 15 min. The measurements were conducted at 37°C. For the recording of dose-response curves the following protocol was used: first, the baseline-BRET was recorded. After ten cycles, 20 µl of agonist-containing solution or external buffer, as a negative control, was added (agonist phase). After a further ten cycles, 20 µl of the saturating concentration of DAMGO or external buffer was added, and 10 more cycles were measured (saturation phase). The measurements were controlled using the software SparkControlTM and were performed as measurements from top. The donor luminescence (nLuc) was recorded at 415–470 nm, and acceptor luminescence (YFP) was recorded at 520–590 nm, respectively. After each measurement, the nLuc and YFP luminescence were exported to Microsoft Excel and the (nLuc/CFP) luminescence ratio was calculated. The negative control trace with just application of external buffer was subtracted and the trace was normalized to the baseline luminescence. For the dose-response curve, the mean of the respective agonist phase was normalized to the saturation phase. Each $n$ indicates one transfection measured as triplicate.

### Statistics and reproducibility
FRET measurements were corrected for photobleaching (using OriginPro 2016). Single-cell FRET measurements were further corrected for background fluorescence, bleed-through of CFP into YFP channel, and false excitation of YFP using Microsoft Excel. Further data analysis was performed with GraphPad Prism 7 (GraphPad Software), as described before[24]. Data are always shown (if not indicated otherwise) as mean ± SEM, and group size is defined as $n$. Statistical analyses were performed with a paired Student's $t$ test or a two-tailed unpaired $t$ test with Welch's correction (as normality of data distribution was not given for every group) or, for more than two groups, by an ordinary one-way ANOVA (as SD's were significantly different, a Brown–Forsythe and Welch's ANOVA test were performed) with Dunnet's T3 multiple comparisons test, as indicated. Differences were considered statistically significant if $P \leq 0.05$. Concentration–response curves were fitted with a nonlinear least-squares fit with variable slope and a constrained top and bottom using the following function:

$$Y = \min + (X^{\text{Hill–slope}}) \times (\max - \min)/(X^{\text{Hill–slope}} + EC_{50}{}^{\text{Hill–slope}})$$

where min and max are the minimal and maximal response and $EC_{50}$ is the half-maximal effective concentration. For the receptor–sensor measurements, the Hill slope was fixed to 1. Inhibition curves were fitted with a nonlinear least-squares fit with variable slope using the following function:

$$Y = \min + (\max - \min)/\left(1 + (IC_{50}/X)^{\text{Hill–slope}}\right)$$

where min and max are the minimal and maximal response, and $IC_{50}$ is the half-maximal inhibitory concentration. The $K_i$-value was calculated using the following function:

$$K_i = IC_{50}/(\{\text{agonist}\}/\{EC_{50}\text{agonist}\})$$

where the concentration of the present agonist is divided by the respective $EC_{50}$ value of the agonist. The kinetics of activation and deactivation were calculated using a one-phase monoexponentially decay nonlinear regression

fit. The following function was used:

$$Y = (Y_0 - NS) * \exp(-K * X) + NS$$

with $Y_0$ being the activation at time zero (in units of the $Y$ axis), NS the nonspecific binding at infinite times (in units of the $Y$ axis) and K the rate constant in inverse units of the $X$ axis. The half-time of activation equals the ln(2) divided by K. For the calculation of the real activation kinetic $K_{on}$ following function was used:

$$K_{on} = EC_{50} / K_{off}$$

using the $EC_{50}$ in M and the $K_{off}$ per minute.

### Reporting summary

Further information on research design is available in the Nature Portfolio Reporting Summary linked to this article.

### Data availability

All data for this study, including the source data used to generate all graphs are available on OSF at https://osf.io/ts8uz/?view_only=b45c7381a6cb465b92127017359821f2. All relevant plasmids generated for this study have been deposited to Addgene (ID numbers 231807-231813).

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

### Acknowledgements

The authors thank Dr. Alwina Bittner for excellent technical assistance in creating cell lines stably expressing the sensors. We thank Dr. Hannes Schihada (University of Marburg, Germany) for the provision of the Go-case BRET-sensor and Dr. Michael Kurz (University of Marburg, Germany) for the support in the development of the FRET based sensors.

### Author contributions

S.B.K., C.Ku., and L.G. performed the experiments and analyzed data. S.B.K., A.L.K., and C.Kr. constructed the plasmids. S.B.K. wrote the manuscript. M.B. supervised the study and edited the manuscript.

## Funding

## Competing interests

The authors declare no competing interests.
