## [Transparent Peer Review file · Communications Biology]

Dynamics of agonist-evoked opioid receptor activation revealed by FRET- and BRET-based opioid receptor conformation sensors

Corresponding Author: Professor Moritz Bünemann

Version 0:

Reviewer comments:

Reviewer #1

(Remarks to the Author)

Interesting paper that creates fluorescence models to study ligand induced opioid receptor conformation changes and receptor-ligand interactions, which are difficult to characterize. More tools are needed in the field.

1. Introduction should have a sentence or two with more basic info on major downstream signaling molecules ("Gi/o proteins or other downstream signaling effectors") before going into the different measurable outcomes (lines 45-48)
2. In line 49, the acronym FRET has a different long form than in line 53 onward; please clarify.
3. Morphine is a full, not partial, signaling agonist. Can the authors comment on why it did not induce a full signal in this model? (line 225+figure 2)
4. Could off rates be compared to drug effect half-life for the clinical drugs? Could the lack of off rate findings be due to absent signaling molecule binding?
5. Does the mono-exponential curve account for receptor crosstalk, which is common among GPCRs—including opioid receptors to other opioid receptors? Could that explain why it did not fully model the kinetics?
6. The conclusion that the lack of downstream signaling proteins creates an unbiased approach (line 392, 436) is too strong, because this could also be interpreted as a weakness to the model. It is hard to interpret the biologic impact of these binding studies since G protein and beta arrestin binding sites are impacted by the fluorophores; their binding impacts receptor conformation/ligand binding.
7. Please clarify in the results and figure legends that KOR/DOR/NOP are human constructs while MOR is rat

Reviewer #2

(Remarks to the Author)

Comments and Suggestions

1. The authors are to be commended for tackling an important problem in pharmacology and collecting a thorough dataset.
2. The authors may also wish to mention early in the manuscript that "the C terminus was truncated, removing most of the phosphorylation sites in the C terminus which are required for arrestin interaction."
3. The authors may wish to mention that the Mu receptor has been expressed as a water soluble protein. See: Xi, J. et al. Characterization of an engineered water-soluble variant of the full-length human mu opioid receptor. *J Biomol Struct Dyn* 38, 4364–4370 (2020). Xi, J. et al. Novel variants of engineered water soluble mu opioid receptors with extensive mutations and removal of cysteines. *Proteins* 89, 1386–1393 (2021).
4. What are the relative advantages/differences between the FRET and BRET reporters?
5. The percent FRET response (~6%), while reproducible and measurable, seems small. How does it compare to other GPCR sensors? If smaller, it is due to the unique distances or angles between the FRET/BRET pairs in the opioid receptors?
6. How do the different kinetics for the different compounds relate to their pharmacology?
7. What might be the advantage of the multiwell format? Small molecule screening?
8. Did the authors test the impact of antibodies?
9. Do these GPCRs ever dimerize?
10. Any novel applications possible for the sensors?

Version 1:

Reviewer comments:

Reviewer #1

(Remarks to the Author)

Authors gave thoughtful and thorough responses to comments. Paper provides useful assays to study opioid receptor conformation, which is a challenge in the field.

Reviewer #2

(Remarks to the Author)

Thank you for considering the reviewers' comments and suggestions in your revised manuscript.

Replies to Reviewers

Reviewer Comments	Responses
Reviewer 1	
1. Introduction should have a sentence or two with more basic info on major downstream signaling molecules (“Gi/o proteins or other downstream signaling effectors”) before going into the different measurable outcomes (lines 45-48)	We thank the reviewer for this suggestion. We have inserted more basic information on the downstream signaling into the introduction (line 31 and lines 45-51).
2. In line 49, the acronym FRET has a different long form than in line 53 onward; please clarify.	Thank you for observing this, we have corrected the acronym FRET to “Förster resonance energy transfer” throughout the whole manuscript.
3. Morphine is a full, not partial, signaling agonist. Can the authors comment on why it did not induce a full signal in this model? (line 225+figure 2)	Thank you for this comment, indeed in clinical context, Morphine is more or less considered as a full agonist. However, several studies showed that on receptor-level Morphine only acts as partial agonist (for example shown by Arden et al, J Neurochem 1995; Johnson et al, Mol. Pharmacol. 2006; McPherson et al, Mol. Pharmacol. 2010). The here developed receptor conformation sensor only displays the activation on receptor level, that’s why Morphine acts as a partial agonist and is not inducing a full signal in this model.
4. Could off rates be compared to drug effect half-life for the clinical drugs? Could the lack of off rate findings be due to absent signaling molecule binding?	We were exploring the time for activation or deactivation of the receptor with our conformation sensor. For the emerging of clinical effects of the drugs, many more signaling steps have an influence. We are not dare to give directly clinical effects based on our sensor. However, the drugs with the clinically longest residency times as methadone and buprenorphine, also showed the slowest off-rate in our system. We also compared the off-rates of our sensor with the ones of the interaction of the receptor with the G-proteins (Supplemental figure 4). Here, the off-rates were slower in comparison to the ones detected by the conformation sensor. This could indicate that indeed the off-rates could be affected by the binding of signaling molecules.
5. Does the mono-exponential curve account for receptor crosstalk, which is	Thank you for this comment, as all experiments were performed in HEK293

common among GPCRs—including opioid receptors to other opioid receptors? Could that explain why it did not fully model the kinetics?	cells, the probability of crosstalk of the sensors with other opioid receptors is highly unlikely, as these cells do not express endogenously other opioid receptors or other potentially crosstalking receptors, but we have not further explored this. Further, the mono-exponential curve could be fully modeled to all the agonists, the only exceptions where the peptide agonists DAMGO and Met-enkephalin. These peptide agonists have potentially also another or respectively larger binding side in the receptor, explaining why we explored a two-phase kinetics for these agonist. In order to enable better comparability between the agonist, we only fitted the first phase of the activation induced by DAMGO and Met-enkephalin. We inserted this explanation also in the manuscript (lines 231 – 234).
6. The conclusion that the lack of downstream signaling proteins creates an unbiased approach (line 392, 436) is too strong, because this could also be interpreted as a weakness to the model. It is hard to interpret the biologic impact of these binding studies since G protein and beta arrestin binding sites are impacted by the fluorophores; their binding impacts receptor conformation/ligand binding.	We thank the reviewer for this comment. We have attenuated our conclusions (lines 409-410 and 458), now using the phrase “potentially unbiased”, as you are right that it’s hard to interpret the influence of the impacted binding sites for G proteins and arrestins.
7. Please clarify in the results and figure legends that KOR/DOR/NOP are human constructs while MOR is rat.	We have inserted the information at several lines in the manuscript (lines 306, 311, 312, 343).
Reviewer 2	
1. The authors are to be commended for tackling an important problem in pharmacology and collecting a thorough dataset.	-
2. The authors may also wish to mention early in the manuscript that "the C terminus was truncated, removing most of the phosphorylation sites in the C terminus which are required for arrestin interaction."	Thank you for this comment, we have inserted the information directly in the introduction in lines 76-78.
3. The authors may wish to mention that the Mu receptor has been expressed as a water soluble protein. See: Xi, J. et al. Characterization of an engineered water-soluble variant of the full-length human mu opioid receptor. J Biomol Struct Dyn 38,	Thank you for the reference to the articles about the expression of the water soluble receptor, we have mentioned them in the discussion (lines 373-376).

4364–4370 (2020). Xi, J. et al. Novel variants of engineered water soluble mu opioid receptors with extensive mutations and removal of cysteines. Proteins 89, 1386–1393 (2021).	
4. What are the relative advantages/differences between the FRET and BRET reporters?	FRET reporter can be used in single-cell approaches under the microscope with a high spatial and temporal resolution or in multiwell format. For FRET, a light source is needed to excite the donor, which induces photobleaching of the donor over time. There is also a spectral overlap of donor and acceptor fluorophores, leading to a necessary correction of false excitation and bleed-through after the measurement. BRET reporter can't be used in microscope setups and give a weaker temporal and spatial resolution. Their advantage instead is the usage in multiwell format, with simple instrument requirements. Further, there is no need of donor excitation via an external light source, leading to a longer stable signal. For BRET, there is also no false excitation or bleed-through and overall in many cases a better signal-to-noise ratio. We have inserted one sentence to the FRET or BRET advantages in the discussion (Lines 423-427).
5. The percent FRET response (~6%), while reproducible and measurable, seems small. How does it compare to other GPCR sensors? If smaller, it is due to the unique distances or angles between the FRET/BRET pairs in the opioid receptors?	Thank you for this comment, the percent FRET response is completely in the range of other established FRET-based receptor sensors varying between 3% (α2A receptor, Rinne et al., PNAS 2013 and TP receptor, Kurz et al, Mol Pharmacol. 2020); 5% (α2A receptor, Schihada et al. 2018); 8% (EP4 receptor, Kurz et al, Mol. Pharmacol. 2023) and up to 12% (PTH receptor, Vilardaga et al, nature biotechnology 2003). Our sensor with a FRET response of 6% is with this in the range of the other established sensors. Every receptor sensor has potentially a unique distance or angle between the FRET/BRET pair, however the exact angle cannot be evaluated with our experimental approach.
6. How do the different kinetics for the different compounds relate to their pharmacology?	See response to reviewer 1 comment 4.

7. What might be the advantage of the multiwell format? Small molecule screening?	The advantage of the multiwell format is indeed the screening of for example small molecules and other applications where a higher throughput is favorable, as the single cell measurements are highly time consuming.
8. Did the authors test the impact of antibodies?	No, we did not test the impact of antibodies, as we saw no implication of them for our tested system.
9. Do these GPCRs ever dimerize?	We did not test dimerization of our sensors, as these has no functional significance in the setup we were using the sensors. Some studies explored the homo-dimerization of the μ opioid receptor. In the study of Möller et al. (Nature chemical biology 2020) it was shown that the μ opioid receptor is exclusively in a monomeric state and if a homo-dimer was formed it was only transiently and had a short lifetime. However, other studies revealed that there are possible hetero-dimers with opioid receptors, for example with the delta opioid receptor and the CB1 receptor (Al-Hasani et al, Anesthesiology 2013). As we performed all experiments in HEK293 cells, heterodimerization should not be observed, as these cells do not endogenously express the regarding GPCRs.
10. Any novel applications possible for the sensors?	Applications for our sensors could not only be the screening for new opioid agonists, but also the determination of the respective activation and deactivation speed of the receptor induced by the new agonist. Further, our sensors could reveal not only the affinity of the agonist to the receptor, further also the efficacy for receptor activation could be determined. We discuss these advantages at different parts of the discussion, including the added sentence in line 459-461.